# Effects of Adding Aquatic-to-Land-Based Physiotherapy Programs for Shoulder Joint Position Sense Rehabilitation

**DOI:** 10.3390/healthcare10020332

**Published:** 2022-02-09

**Authors:** Alexandra Camelia Gliga, Nicolae Emilian Neagu, Horatiu Valeriu Popoviciu, Tiberiu Bataga

**Affiliations:** 1Institution Organizing University Doctoral Studies of George Emil Palade University of Medicine, Pharmacy, Science, and Technology of Targu Mures, 540139 Targu Mures, Romania; tbataga@gmail.com; 2Laboratory of Rehabilitation, Physical Medicine and Balneology, County Clinical Emergency Hospital of Targu Mures, 540136 Targu Mures, Romania; 3Department of Functional and Complementary Sciences, Faculty of Medicine, George Emil Palade University of Medicine, Pharmacy, Science, and Technology of Targu Mures, 540139 Targu Mures, Romania; neagutgm@gmail.com; 4Department of Clinical Sciences and Internal Medicine, Faculty of Medicine, George Emil Palade University of Medicine, Pharmacy, Science, and Technology of Targu Mures, 540139 Targu Mures, Romania; horatiu.popoviciu@umfst.ro; 5Clinic of Rheumatology, County Clinical Emergency Hospital of Targu Mures, 540136 Targu Mures, Romania; 6Department of Clinical Sciences, Faculty of Medicine, George Emil Palade University of Medicine, Pharmacy, Science, and Technology of Targu Mures, 540139 Targu Mures, Romania; 7Clinic of Orthopaedics and Traumatology, County Clinical Emergency Hospital of Targu Mures, 540136 Targu Mures, Romania

**Keywords:** rehabilitation, physiotherapy, proprioception, joint position sense, shoulder joint

## Abstract

There is limited evidence regarding the effects of aquatic-based physiotherapy on shoulder proprioception following post-traumatic injury to the joint. The main aim of this study was to investigate the effects of additional aquatic-based rehabilitation to a land-based physiotherapy program on shoulder joint position sense (JPS) rehabilitation. Forty-four individuals (mean age 44.50 ± 10.11) who had suffered a post-traumatic shoulder injury less than five months previously were pseudo-randomly allocated equally into a control group (9 females, 13 males) and experimental group (6 females, 16 males). Both groups received individualized standard land-based physiotherapy on average for 50 min per session, with five sessions per week for four consecutive weeks. The experimental group received an additional 30 min of personalized aquatic-based therapy during each session. Shoulder JPS was assessed by flexion (60°), extension (25°), abduction (60°), internal rotation (35°) and external rotation (35°) positions prior, halfway through, and after the intervention. Shoulder JPS improved significantly for all positions for both the control group (*p* < 0.03) and the experimental group (*p* < 0.01). No significant differences between the control group and the experimental group were found for change in shoulder JPS over time. Our results indicate that shoulder JPS can be significantly improved among individuals with post-traumatic injury to the joint through four weeks of personalized physiotherapy. The addition of aquatic-based exercises to standard land-based therapy did not, however, show significant benefits, and thus cannot be recommended for the improvement of shoulder JPS based on our findings.

## 1. Introduction

The subject of proprioception lies at the border between neurophysiology and neuropsychology. Sir Charles Sherrington introduced for the first time the term “proprioception”, describing the perception of the movement but also the position of the body in space, in the absence of visual feedback [1]. Simplifying Sherrington’s proposed distinction, ”perception” refers to experiencing the environment around the body, while ”proprioception” is an experience of the body itself [2]. The literature data consider proprioception as the sum of multiple neurological feedbacks from many systems of the human body, with a role in functioning and adapting motor control [3,4]. Over the years, the concept of proprioception has evolved into a general topic of major interest, which includes kinaesthesia (the sense and awareness of segmental and joint passive and active movements); somesthesia (the sense of segmental and articular passive or active positioning—JPS); active or passive reproduction of the movements; detection of vibrations and muscle strength [5,6].

Given the importance of proprioception on performing movements in a coordinated manner, we can see a strong association between various musculoskeletal disorders and their negative influence on proprioception [7,8,9,10]. When it comes to the shoulder joint, due to the high degree of mobility, the synchronization of active and passive structures has a major role in its stability [11,12,13]. For this reason, shoulder neuromuscular control plays an important role by coordinating muscle activity during voluntary functional movements, which involve the co-activation of the shoulder muscles, reflexes, and the regulation of strength and endurance [14]. The impairment of proprioception in the shoulder joint is caused by the lesions of capsule-ligament structures, which lead to decreased stimulation of the mechanoreceptors. Pacini corpuscles, Ruffini endings, Golgi tendon organs, and neuromuscular spindles have been identified in the shoulder joint [15,16]. Of the shoulder conditions which cause major deficits in proprioception, researchers found joint instability, impingement, rotator cuff tears, frozen shoulder, proximal humerus or distal fractures of the clavicle, and acromioclavicular dislocations [8,17,18,19,20,21]. Impaired proprioception, due to joint instability, is also correlated with joint rhythm disorders and neuromuscular disorders [8].

When it comes to shoulder rehabilitation, programs have been designed to restore motor function and to reduce the risk of re-injury, with the help of physiotherapy, hydrotherapy, thermotherapy, electrotherapy, manual therapy or mirror therapy [22,23]. In addition to the rehabilitation of motion range, or muscle strength and endurance, proprioception is the function that optimizes the accuracy of movements, thus playing an important role in the motor control, which promotes and even determines the accuracy of fine motor skills at the level of the injured upper limb [23,24]. The effects of active movement exercise on shoulder proprioception rehabilitation, after different injuries, are not completely understood. Nonetheless, Casadio et al. showed that active exercises can increase both the functions of the upper limb and the level of proprioception by: (1) making the patient focus more on joint positioning during active movements; and (2) increasing the sensitivity of the neuromuscular spindles [25]. Therapeutic exercise programs lead to the improvement in the joint position sense (JPS) by increasing peripheral and central neuronal adaptation [26]. Following these rehabilitation programs, the Golgi tendon organs become desensitized, a phenomenon that leads to an increase in the sensitivity of the neuromuscular spindles, which has been shown to improve proprioception [27].

The beneficial effect of closed kinetic chain exercises on rotator cuff and scapula-thoracic muscles improved JPS reproduction but did not show statistically significant differences when compared to open kinetic chain exercises in a group of thirty-six healthy subjects, as indicated in Lin and Karduna research [19]. Spargoli found no difference between eccentric and concentric exercises, in terms of their impact on the rehabilitation of proprioception [28]. Improvements in proprioception, by increasing the sensitivity of the Golgi tendon organ and the affected neuromuscular spindles through active resistive exercises, were indicated by Miall et al. [29]. Various unstable surfaces are frequently used in the training and rehabilitation of proprioception, with the role of increasing the somatosensory ability and improving reflex responses [15]. It has also been found that the sensory information received by the visual channel (for example mirror therapy) and the tactile channels play an important role in the rehabilitation of proprioception [23]. Neuromuscular electrical stimulation and biofeedback have been also used in proprioception rehabilitation.

There is limited evidence regarding the effects of aquatic-based physiotherapy on shoulder proprioception following post-traumatic injury to the joint. Previous studies indicate that aquatic-based rehabilitation programs improved the shoulder’s range of motion and strength, diminished pain and restored functional movement patterns [30,31,32,33]. One of the mechanical effects of water is the relative decrease in body weight determined by the Archimedes’ strength and depth of immersion. Other biomechanical effects are caused by hydrostatic pressure and water viscosity, which plays a role in drainage, circulation and sensory stimulation, with proprioceptive, somesthetic, and kinesthetic facilitating effects [34,35]. Last but not least, one other mechanical effect of water is its influence on the speed of movement (significantly diminished) and the opposite resistance in performing these movements (significantly increased). Due to these two, sensory feedback increases, the body awareness sense is promoted [36]. Additionally, the opposite resistance property of the water facilitates active resistive exercises, which are well known for improving shoulder JPS [29]. When it comes to the thermal effect, warm water plays an important role in the somatosensory stimulation training, when applied 30 min per session, five sessions per week for six consecutive weeks, as shown in the research of Chen et al. on a total of thirty-six acute stroke patients [37]. On the other hand, Khanmohammadi et al. concluded that a single fifteen-minute cold-water immersion at 6 °C does not affect or improve ankle JPS in thirty healthy female participants [38].

The aims of this study were to (1) investigate the effects of personalized physiotherapy on shoulder JPS among individuals with post-traumatic injury to the joint and (2) investigate whether additional aquatic-based therapy would have significant effects on shoulder JPS compared with only land-based rehabilitation. We hypothesized that personalized physiotherapy would significantly improve shoulder JPS and that additional aquatic therapy would improve shoulder JPS significantly more than land-based therapy alone.

## 2. Materials and Methods

### 2.1. Study Design

This was a single-center, prospective, randomized trial, conducted in Targu Mures, Romania. During a period of eleven months, between October 2020 and September 2021, participants were included in individualized and personalized land- and aquatic-based rehabilitation programs along with various baseline, intermediate and post-intervention assessments. The study was approved by the Ethics Committee of the Country Clinical Emergency Hospital of Targu Mures (Decision no. Ad. 23666 from 16 October 2020) and by the Ethics Committee of the “George Emil Palade” University of Medicine, Pharmacy, Science and Technology of Targu Mures (Decision no. 1149 from 15 October 2020).

### 2.2. Participants

Forty-four participants, former outpatients of the Clinic of Orthopaedics and Traumatology, County Clinical Emergency Hospital of Targu Mures, took part voluntarily in the personalized physiotherapy programs and in the assessments related to the present research, after being contacted by the investigators. Before inclusion, all participants signed an informed consent. The number of included participants was limited due to the current pandemic context. All participants presented recent history of non-operated post-traumatic shoulder conditions, such as: humeral proximal epiphysis fracture, clavicle distal extremity fracture and acromioclavicular dislocations.

In addition to the existence of post-traumatic conditions, the inclusion criteria consisted of: (1) less than five months passed since the traumatic injury; (2) injury occurred on the dominant upper limb; (3) range of motion re-established in proportion of approximately 70%, with painless active mobilizations of the shoulder possible in all directions, with values of minimum 120° flexion and abduction, 40° extension, 50° internal and external rotations; (4) possibility of classification, objectified by testing, of the participant at the end of the second, or at the beginning of the third, stage of rehabilitation, namely at the end of the stage when regaining full active range of motion or at the beginning of the stage of rehabilitation of muscle strength and motor control; (5) moderate physical activity level; and; (6) age between 20 and 60 years. The exclusion criteria were: (1) history of associated conditions, such as neurological, psychiatric, inflammatory, rheumatic, severe cardiovascular, infectious or diabetes; (2) previous orthopaedic surgery within the last six months; (3) undergoing drug treatment that could influence the outcome of this research; (4) hydrophobia; (5) sedentary or vigorous physical activity level; and (6) undergoing other procedures, such as electrotherapy or heat therapy. Participants were informed that, during this research, they should report any changes to the medication used and any other physical activities performed in addition to the physiotherapy sessions. They were also informed that they could withdraw their participation at any time during the study.

### 2.3. Study Samples

Following the presentation of the agreement and signing of the informed consent, the participants were pseudo-randomly allocated equally into one of the two groups of our research: the experimental and the control group. Randomization was performed using a digital program, and both groups had similar characteristics, established by pre-testing a series of correlative parameters, such as: the same number of participants, pathology, anthropometric characteristics and manual dominance.

### 2.4. Study Settings

Our prospective study was conducted within the Department of Movement Sciences and the Alma Mater Rehabilitation Centre of “George Emil Palade” University of Medicine, Pharmacy, Science, and Technology of Targu Mures and within the Laboratory of Rehabilitation, Physical Medicine and Balneology from the County Clinical Emergency Hospital of Targu Mures, Romania.

### 2.5. Interventions

All the participants were included in individual medical rehabilitation programs, for a period of four weeks, five days a week, conducted by the same physiotherapist. The control group (*n* = 22 participants) benefited from individualized and personalized physiotherapy programs which included the use of assisted, active, or resistive mobilizations, neuroproprioceptive facilitation techniques, proprioception, and motor control exercises, stretching, and muscle relaxation methods. The average duration of a physiotherapy session was 50 min, structured in five parts: (1) warm-up with passive, assisted or active exercises (approx. 15 min); (2) resistive active exercises for strength rehabilitation (approx. 10 min); (3) proprioception and motor control rehabilitation exercises (approx. 20 min); and (4) stretching (approx. 5 min) and (5) relaxation (approx. 5 min).

In addition to the contents, methods, and techniques applied to the control group, the experimental group (*n* = 22 participants) benefited from individualized and personalized hydrotherapy sessions, performed immediately after the end of the physiotherapy sessions described above. We mention the fact that the last two parts of the physiotherapy programs were only applied at the end of the hydrotherapy session; thus, the experimental group benefited from an approximately 50 min land-based physiotherapy session, followed by an approximately 30 min hydrotherapy session, structured in three parts: (1) warm-up with the role of adjusting to the aquatic environment (approx. 5 min); (2) specific water-based exercises for proprioception rehabilitation (approx. 20 min); and (3) post work-out recovery and relaxation (approx. 5 min). The pools in which these rehabilitation sessions took place are located in the Laboratory of Rehabilitation, Physical Medicine and Balneology from the County Clinical Emergency Hospital of Targu Mures, Romania. The water temperature in the pool was 30–32 °C, and the ambient temperature was 27–28 °C. The water we used was from the general supply network and it had no special biochemical properties. The dimensions of the two pools were 8 × 8 m, with a depth of 1.8 m, and 6 × 6 m, with a depth of 1.6 m.

Even though, in terms of the total duration, the therapy sessions applied to the experimental group were longer than the therapy sessions applied the control group, we tried to monitor our intervention in such a way that all of the main parameters of the experiment applied to the participants, respectively; the intensity and the number of therapeutic exercises instrumented were almost similar for both groups. We may, therefore, state that the action of the independent variable on the groups was symmetrical.

The experimental curve was a progressive curve, and the therapeutic density was directed and monitored by controlling the duration of the breaks, by monitoring the evolution of the intensity and difficulty of the exercises, by performing the exercises with or without visual control, etc. As an example, we shall describe a small part of the rehabilitation exercise package, which was individualized and adapted to each participant. The individualization of the rehabilitation programs is the reason why we do not present the number of repetition or series performed by the participants. Of note, the exercises performed were not limited only to those described in Table 1 and Table 2 below.

### 2.6. Joint Position Sense Assessment

The assessments of the participants were performed three times: at the beginning of the rehabilitation program (baseline assessment); a sequential assessment applied at the end of the second week of rehabilitation, performed as an indicator of efficiency and regulator of the physiotherapy programs, of which results were not used for analysis in the current study (intermediate assessment); and at the end of the four weeks of land- and aquatic-based rehabilitation programs (post-intervention assessment).

For the assessment of the JPS, we used the Kinesimeter, a precise angular motion-measuring device, designed in the M2 Department of Functional and Complementary Sciences, Department of Movement Sciences, Faculty of Medicine, “George Emil Palade” University of Medicine, Pharmacy, Science, and Technology of Targu Mures, Romania. The Kinesimeter consists of a vertical stable stand with a horizontal movable arm, which can rotate around an axis. This device can be connected to a laptop, and based on a software interface, it can reproduce the movements of the mobile arm in the form of a real-time oscillogram, providing detailed numerical data related to amplitude, duration, frequency, chronology, and shape. Numeric data were generated in Excel format.

In a previous study, we showed lower inter-tester differences and statistically significant improvements in the error rate, when the Kinesimeter was used for assessing the shoulder’s range of motion, compared to the classical goniometer [39]. From these results, we have concluded that our device is a valid and reliable instrument to measure the precise angles and degrees of the shoulder joint.

The method used for assessing the JPS was the passive positioning of the upper limb with active reproduction of the previous movement, both without visual control. The upper limb was passively positioned at the same reference angle for all the participants, as follows: 60° for flexion and abduction; 35° for internal and external rotation; and 25° for extension. After passively reaching the indicated position, each participant was allowed to focus on the reference angle for no longer than 10 s, before the upper limb was returned to the anatomical relaxed position. Subsequently, the participant was asked to reposition his/her upper limb up to where she/he believed that the reference angle was previously. The difference between the reference angle and the indicated angle was measured in three consecutive repetitions for every shoulder movement assessed. The average mean of these three error values was recorded as the absolute error.

During the evaluations, under proprioceptive control, all participants wore eye masks that prevented peripheral view of the arm. The measurements were performed from the initial seated position, with the upper limb in anatomical position, elbow in extension and hand in supination (for shoulder JPS assessment during the movement of flexion, extension, and abduction), and from the lying position, the shoulder joint being in 90° abduction, the hand in neutral position, and the elbow in 90° flexion (for shoulder JPS assessment during the movement of internal and external rotation). In all the measurements performed, the positioning of the axis, the fixed and the mobile arm of the Kinesimeter complied with the general rules specific to the range of motion measurements (the axes of rotation between the Kinesimeter and the shoulder joint being concentric).

### 2.7. Statistical Methods

The statistical analysis included elements of descriptive statistics (frequency, percentage, mean, median, and standard deviation) and elements of inferential statistics. The Shapiro–Wilk test was applied to determine the distribution of the analyzed data series. All of the analyzed data series had Gaussian distributions. For comparison of the means, a Student’s *t*-test was used as well as a Student’s *t*-test with Welch’s correction, respectively. For intra-group comparisons, a paired Student’s *t*-test was applied. For inter-group comparisons, an unpaired Student’s *t*-test for unpaired data, and unpaired Student’s *t*-test with Welch’s correction for unpaired data, were applied where the variances were significantly different. The Mann–Whitney test was used to compare the differences between the median values of the difference between the final and the initial values of the two independent groups. The post hoc power analysis sample size for two independent study groups with continuous outcomes was applied. For multiple comparisons corrections, the Bonferroni correction for multiple testing was applied and we found no significant statistical difference between the values collected in the three consecutive repetitions. The effect size was represented by eta squared (η^2^) to show the proportion that can be attributed to independent variable from the total variation in the dependent variable. Eta squared was calculated using one-way ANOVA, univariate analysis of variance. Eta squared values of 1 to 6% (η^2^ = 0.01) indicated small effects, values of 6 to 14% (η^2^ = 0.06) medium effects, and values of 14% (η^2^ = 0.14) large effects. The significance threshold chosen for *p* was 0.05. Statistical analysis was performed using the GraphPad Prism for Windows (trial version).

## 3. Results

The post hoc analysis indicated that this sample size yielded a power of 72.2% with the probability of type I error of 0.05.

### 3.1. Anthropometric Characteristics

The anthropometric characteristics of the investigated participants, as well as their age, sex, manual dominance, and post-traumatic condition are presented in Table 3 and Table 4. The mean values of age, height, body weight, and body mass index were not statistically significantly different between the control and experimental group (Table 3).

In the control group, there were 40.91% female participants and 59.09% male participants, and in the experimental group, there were 27.27% female participants and 72.73% male participants. Based on hand dominance, 9.09% were left-handed and 90.91% were right-handed in the control group; 18.18% were left-handed and 81.82% were right-handed in the experimental group. Of the participants included in the control group, 59.09% presented a recent history of proximal humerus fracture, and 54.54% of the participants included in the experimental group shared the same condition. The distal clavicle fracture was a recent condition in 18.18% of the participants from the control group and in 31.82% of the participants from experimental group. The acromioclavicular dislocations were recent in 22.73% of the participants from the control group and in 13.64% of the participants from experimental group. All data regarding gender, manual dominance and injury distribution are detailed in Table 4.

### 3.2. Joint Position Sense Comparative Results

In both the control and experimental group, the post-intervention results were statistically significant when lower than the baseline assessment results for all studied categories (flexion, extension, abduction, internal and external rotations). When performing the inter-group comparative analysis, the baseline shoulder JPS values were not statistically significant between the control and experimental groups. The post-intervention JPS mean values for the measurement of flexion, abduction, internal and external rotation were statistically significantly lower between the two groups. No statistical significance was shown regarding the post-intervention JPS for extension parameters, Table 5.

The effect size was calculated and we identified the following (summarized in Table 5): flexion—13.4% from value variation is determined by the type of intervention, extension—2.2%, abduction—11.6%, internal rotation—9.1% and external rotation—12.8%. For flexion, abduction, internal and external rotations, medium effects were indicated, while for extension, small effects were indicated.

The differences of the dependent variables between the baseline and post-intervention assessment were compared between the control and experimental group and no statistically significant difference was found: all *p* > 0.05 (Table 6).

## 4. Discussion

### 4.1. Aquatic- and Land-Based Rehabilitation

The aims of this study were to (1) investigate the effects of personalized physiotherapy on shoulder JPS among individuals with post-traumatic injury to the joint; and (2) investigate whether additional aquatic-based therapy would have significant effects on shoulder JPS compared with only land-based rehabilitation. It was found that: (1) shoulder JPS improved significantly for all positions for both the control group (*p* < 0.03) and the experimental group (*p* < 0.01); and (2) no significant differences between the control group and the experimental group were found for the change in shoulder JPS over time. Our hypothesis that personalized physiotherapy significantly improves shoulder JPS is confirmed, while the hypothesis that additional aquatic therapy improves shoulder JPS significantly more than land-based therapy alone is not confirmed.

Taking into account the design of this study, care should be taken when interpreting the results. Even though, in terms of the total duration, the therapy sessions applied to the experimental group were longer than the therapy sessions applied the control group, we tried to monitor our intervention in such a way that all of the main parameters of the experiment applied to participants were instrumented almost similar for both groups.

If the influence of aquatic-based therapy on shoulder rehabilitation is considered, our results are similar to those of Brady et al., in which 18 subjects with rotator cuff repairs were treated over a period of 12 weeks [30]. This research indicates significant improvements in both the aquatic- and land-based groups, and concludes that a combined aquatic- and land-based program is a feasible and viable alternative to the classical land-based exercise rehabilitation program. In recent years, the popularity of aquatic therapy has increased [34], but the influence of aquatic training on JPS is still not well-documented. A recent meta-analysis on individuals with hip or knee replacement shows better results in muscle strength and the active range of motion rehabilitation when aquatic therapy is used, compared to protocols that included only classical land therapy [40].

### 4.2. Joint Position Sense Assessment

Assessing the proprioceptive function can be a real challenge, even nowadays, because of the lack of a gold-standard test [41]. Various techniques to assess different sub-modalities proprioceptive mechanisms have been reported in the literature, but the three main techniques are: threshold to the detection of passive movement; joint position reproduction; and active movement extent discrimination assessment [42,43].

In the present study, the joint position reproduction technique was used, with the passive positioning of the upper limb and active reproduction of the previous movement without visual control. For this research, the participants were assessed from a seated and lying position, with the help of the Kinesimeter.

For our research, we followed the methodology of other studies, taking into account the absolute error, and not constant error [42,44]. In some research, shoulder proprioception assessment strictly investigated the movement of rotation [45,46], while others investigated only flexion and abduction movements [47,48]. For this reason, we assessed flexion and extension, abduction, internal and external rotations. Research on shoulder and knee JPS indicates that statistically larger errors are found while the active reproduction is performed compared with passive reproduction of joint position. [42,49] Regarding the positioning of the participant when assessing shoulder proprioception, it has been found that there are no statistically significant differences in outcome, regardless of whether the participant is in a supine, sitting, or standing position [50,51].

### 4.3. Limitations of the Study

The present study has some limitations to consider. First, the design of our study makes it hard to ascertain whether the post-intervention results of the experimental group are influenced by the general benefits of hydrokinetic therapy or by longer therapy sessions. This is caused by the absence of a control condition accounting for the duration of therapy. Second, the lack of a standardized JPS assessment tests and the lack of information about the reliability and validity of the Kinesimeter. Third, the number of included participants was limited due to the current pandemic context. Fourth, although all participants presented a recent history of non-operated post-traumatic shoulder conditions, the variation of these three pathologies (humeral proximal epiphysis fracture, clavicle’s distal extremity fracture and acromioclavicular dislocations) does not allow us to draw conclusions that are specific for each of these conditions. The confirmation of the results obtained by us requires additional investigations, functional tests and the performing of assessments at different levels of the range of motion of the shoulder. In the future, we aim to investigate a similar hypothesis, on higher study groups with post-traumatic shoulder conditions, but with several methodology changes. Moreover, we will also compare the results of land-based rehabilitation with those obtained from isolated aquatic intervention, observing the control conditions mentioned above.

## 5. Conclusions

The findings of this study indicate that shoulder JPS can be significantly improved after only four weeks of personalized physiotherapy among individuals with post-traumatic injury to the joint. However, the lack of significant benefits from additional aquatic-based therapy indicates the lack of value for shoulder JPS for such exercises. Further research is required to assess whether aquatic-based therapy can provide benefits over land-based therapy on other relevant outcomes such as muscular strength and the joint range of motion.

## Figures and Tables

**Table 1 healthcare-10-00332-t001:** Indicative example of the set of exercises used in the physiotherapy session.

The Structure of a Physiotherapy Session	Exemplification and Description of the Exercises
Warm-up with assisted or active exercises (approx. 15 min)	From supine starting position: 1. Performing the abduction movement of the shoulder for the entire range of motion, with the help of free suspension therapy; 2. Assisted exercises to promote mobility, stability, and controlled mobility, performed in all directions of the shoulder movement (flexion, extension, abduction, adduction, internal and external rotation), characteristic of neuroproprioceptive facilitation techniques. From sitting starting position: 3. Performing the movement of shoulder flexion for the entire range of motion with the help of a pulley therapy system. From standing starting position: 4. Passing a small ball from one hand to the other around the pelvis, at different amplitudes of movement; 5. Performing the sagittal plane circumduction of the shoulder, the participant being positioned laterally to a fixed wheel axis, located in the central point of the shoulder joint; etc.
Resistive active exercises for strength rehabilitation (approx. 10 min)	From prone lying starting position: 1. With the shoulder positioned at the edge of the bed and the upper limb hanging, the scapula retraction is performed with the isometric maintenance of the position; 2. From the same position as exercise no. 1, performing shoulder flexion, followed by extension against external resistance applied through a sandbag located at the distal extremity of the forearm. The working method was achieved through progressive resistance fractional technique (De Lorme–Watkins method). From sitting starting position: 3. Moving the shoulder against the resistance of an elastic band in internal and external rotation; 4. Mobilization of the upper limb in abduction, against external resistance applied by a weight-type mechanotherapy system, etc.
Proprioception and motor control rehabilitation exercises (approx. 20 min)	From supine starting position: 1. With 90° flexion of the shoulder, extended elbow, pronated and extended hand, the patient presses a Bobath^®^ ball against the wall and performs rapid flexion and extensions of the elbow, with the detachments of the ball from the palm, without it falling and without visual control. From side lying starting position: 2. Regardless of the position of the shoulder, the elbow positioned on the torso in 90° flexion, the hand in pronation and holding a Bodyblade^®^ oscillating device parallel to the floor, the participant performs rapid movements of internal and external rotation, overcoming the oscillations and the resistance, without visual control. From prone lying starting position: 3. The participant advances so that the pelvis is located at the end of the work table, the torso being supported by placing both hands on a Bobath^®^ ball, with extended elbows, and 90° shoulder flexion. From this position, the ball is slightly rolled forward–backward or left–right, while maintaining the correct spine position, without visual control. From prone kneeling starting position: 4. With both palms positioned on the reversed Bosu^®^ ball, the participant performs regular push-ups. From sitting starting position: 5. With the help of the Kinesimeter device, used both for training and evaluating the proprioception of the shoulder, the passive positioning of the upper limb is performed, with the active reproduction of the previous movement, in all directions of movement, without visual control. From standing starting position: 6. Throwing a rubber ball the size of a handball against the wall and catching it from a distance of 20 cm, without changing the amplitude of the shoulder joint, which is in 90° abduction, maximum external rotation and 90° flexion of the elbow; 7. “Drawing” numbers on the wall without visual control, with a compressed Bobath^®^ ball, the participant being positioned laterally, shoulder 90° abducted and extended elbow; etc.
Stretching (approx. 5 min)	Auto-passive or passive stretching of the deltoid, biceps, triceps, coracobrachial, teres minor and major, infraspinatus, supraspinatus, rhomboid, pectoral, trapezius, latissimus dorsi, anterior serratus and levator scapulae muscles.
Relaxation (approx. 5 min)	Jacobson’s method of progressive muscle relaxation, applied on different muscle groups, with 7 s of isometric contraction, followed by 20 s of relaxation.

**Table 2 healthcare-10-00332-t002:** Indicative example of the set of exercise used in the hydrotherapy session.

The Structure of a Hydrotherapy Session	Exemplification and Description of the Exercises
Warm-up with the role of accommodation in the aquatic environment (approx. 5 min)	From floating supine starting position: 1. With one pool noodle positioned under the knees, one positioned at the level of the lumbar spine and one slim cervical floating belt, the participant performs free abduction of both upper limbs for full range of motion of the shoulder on the surface of the water. From standing starting position: 2. Free underwater circumduction is performed without the upper limb exceeding the water surface; 3. With both hands positioned in the middle of a pool noodle, floating on the water surface, shoulders positioned in 90° flexion and elbows extended, the pool noodle is immersed; 4. With both hands wearing aquatic gloves and elbows positioned in 90° flexion, close to the trunk, the external rotation of the shoulder is performed simultaneously, overcoming the resistance of the water, followed by internal rotation; etc.
Specific water-based exercises for proprioception rehabilitation (approx. 20 min)	From floating supine starting position: 1. With one pool noodle positioned under the knees, one positioned at the level of the lumbar spine, one slim cervical floating belt, and wearing two aquatic gloves, the participant performs fast simultaneous movements of splashing the water through flexing and extending the shoulder joint of both upper limbs; at the same time, without visual control, the participant slowly moves both upper limbs in abduction and adduction, on the water surface, and tries to maintain the position of the trunk by avoiding possible imbalances; 2. Without any support from assistive floating devices, but with both shoulder joints positioned in 90° abduction and elbows extended, the participant rotates their trunk at a brisk pace, in both directions, keeping the upper limbs in isometric contractions, so that their position is not disturbed by the movement itself or by the water waves, without visual control. From floating prone lying starting position: 3. With the help of a pool noodle positioned at the level of the pelvis, both upper limbs are positioned in maximum flexion of the shoulder joint and grip the support bar of the pool. Alternately, without visual control, the participant will detach a limb and mobilize it so that the palm will touch the opposite shoulder; subsequently, the same limb will be returned to the fixed bar and mobilized to touch the opposite hip joint; to avoid hyperextension of the cervical spine, it is recommended to use a professional snorkeling mask. From sitting starting position: 4. With the help of a portable water-resistant laser device held in the participant’s hand, the shoulder joint being positioned in 90° flexion, the geometric shapes on the edge of the pool are reproduced, with or without visual control, at different levels of joint amplitude, while the participant is positioned 4 m away from the target area. From standing starting position: 5. Alternative immersion of a pool noodle, starting from the surface of the water, with the left upper limb, then with the right one, keeping the elbow extended and going in the direction of the opposite hip; the exercises will be performed in a jerky rhythm, with rapid mobilizations of the upper limb, following the model of the neuroproprioceptive facilitation technique called repeated contractions, without visual control; 6. “Drawing” numbers without visual control, with an aquatic dumbbell held in the participant’s hand, while the shoulder is positioned in 90° abduction and the elbow is extended; 7. With the help of two aquatic dumbbells, one in each hand, the alternative circumduction of the limbs is performed on the entire possible range of motion, exceeding the water surface without visual control, and the rapid change of the direction of movement is performed when hearing the sound signal given by the physiotherapist, etc.
Post work-out recovery and relaxation (approx. 5 min)	From standing starting position: 1. Free underwater circumduction is performed, without the upper limb exceeding the water surface; 2. Simultaneous free mobilization of both upper limbs in the form of breaststroke technique of swimming. From sitting starting position: Various breathing exercises are performed for relaxation, combined with the mobilization of the upper limbs, etc.

**Table 3 healthcare-10-00332-t003:** Distribution of the participants according to age and anthropometric characteristics.

Participants Characteristics	Control Group Mean ± SD	Experimental Group Mean ± SD	*p* Value
Age (years)	44.50 ± 10.11	48.05 ± 9.02	0.226
Height (cm)	173.10 ± 7.83	170.80 ± 7.54	0.323
Weight (kg)	78.05 ± 10.47	74.36 ± 8.43	0.206
BMI (kg/m^2^)	26.06 ± 2.79	25.51 ± 2.18	0.477

**Table 4 healthcare-10-00332-t004:** Distribution of the participants according to sex, manual dominance, and post-traumatic condition.

Participants Characteristics	Control Group		Experimental Group	
	No.	%	No.	%
Sex (female/male)	9/13	40.91/59.09	6/16	27.27/72.73
Hand dominance (left/right)	2/20	9.09/90.91	4/18	18.18/81.82
Proximal humeral fracture	13	59.09	12	54.54
Distal clavicle fracture	4	18.18	7	31.82
Acromioclavicular dislocation	5	22.73	3	13.64

**Table 5 healthcare-10-00332-t005:** Comparative analysis of the initial and final results of the control group.

		Control Group Mean ± SD (Median)	Experimental Group Mean ± SD (Median)	*p* Value	Effect Size
Flexion (60°)	Baseline assessment results	4.80 ± 2.11 (4.33)	4.33 ± 1.85 (3.84)	0.437	
	Post-intervention assessment results	3.32 ± 1.04 (3.50)	2.53 ± 1.01 (2.33)	0.015	13.4%
*p* value		0.010	<0.001		
Extension (25°)	Baseline assessment results	6.82 ± 1.82 (6.33)	6.36 ± 2.63 (5.67)	0.511	
	Post-intervention assessment results	5.43 ± 2.58 (5.17)	4.79 ± 1.72 (5.33)	0.341	2.2%
*p* value		0.001	0.002		
Abduction (60°)	Baseline assessment results	5.93 ± 3.14 (6.17)	5.18 ± 2.73 (5.33)	0.407	
	Post-intervention assessment results	3.95 ± 2.47 (3.84)	2.53 ± 1.42 (2.33)	0.025	11.6%
*p* value		0.001	<0.001		
Internal rotation (35°)	Baseline assessment results	6.55 ± 1.59 (6.84)	6.09 ± 2.38 (6.00)	0.460	
	Post-intervention assessment results	5.51 ± 1.26 (5.67)	4.45 ± 2.07 (4.00)	0.048	9.1%
*p* value		0.022	0.004		
External rotation (35°)	Baseline assessment results	6.89 ± 2.44 (6.84)	6.15 ± 2.11 (6.00)	0.287	
	Post-intervention assessment results	6.15 ± 1.81 (6.17)	5.02 ± 1.14 (5.00)	0.018	12.8%
*p* value		0.027	0.002		

**Table 6 healthcare-10-00332-t006:** Comparative analysis of the change in shoulder JPS over time for the control and experimental group.

Post-Intervention— Baseline Assessment Results	Control Group Mean ± SD (Median)	Experimental Group Mean ± SD (Median)	*p* Value
Flexion (60°)	−1.49 ± 2.45 (−0.84)	−1.80 ± 1.94 (−1.33)	0.384
Extension (25°)	−1.39 ± 1.68 (−1.33)	−1.58 ± 2.04 (−1.17)	0.747
Abduction (60°)	−1.97 ± 2.41 (−1.67)	−2.65 ± 2.20 (−2.17)	0.306
Internal rotation (35°)	−1.03 ± 1.95 (−0.84)	−1.64 ± 2.40 (−1.67)	0.363
External rotation (35°)	−0.74 ± 1.47 (−0.50)	−1.14 ± 1.53 (−1.33)	0.389

## Data Availability

The data presented in this study are available on request from the corresponding author.

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
