# Peer review of "Effects of Adding Aquatic-to-Land-Based Physiotherapy Programs for Shoulder Joint Position Sense Rehabilitation"

_healthcare, 2022, doi:10.3390/healthcare10020332_

Round 1
Reviewer 1 Report
I recommend the publication in this form. However, I have some questions and recommendations.
The introduction is appropriate and it describes the relationship between joint position and the nervous system.
The description of the exercises is good, although it might be advisable to include pictures of them.
Is it possible to relate proprioception to other outcomes such as pain or specific shoulder assessment scales?
In the future, would it be possible to compare the results with an isolated aquatic intervention?
Thank you.
Author Response
Esteemed Editor and Reviewer from Healthcare Journal,
Thank you for giving us the opportunity to submit a revised draft of the manuscript. Please see the attached point-by-point response to Reviewer #1 suggestions and comments.
Alexandra – Camelia Gliga

Reviewer 2 Report
A brief summary
The aim of this paper was to investigate the differences between land-based rehabilitation and additional aquatic-based rehabilitation programmes on shoulder joint position sense (JPS) among individuals with shoulder conditions. The authors have succeeded in completing four-week rehabilitation programmes among 44 individuals who overall show improvement in shoulder JPS. This is commendable given that the intervention was also completed during the current pandemic. Some of the exercises performed during the intervention are described and are interesting for the reader. Based on the significant differences between the control group and the experimental group post-intervention, which were not evident pre-intervention, the study encourages the use of additional aquatic-based rehabilitation among this population.
General concept comments
The study design is ultimately flawed, as it seems the individuals in the experimental group received a total of 10 hours extra supervised rehabilitation compared to those in the control group. It is therefore impossible to ascertain whether any benefits of the type of exercise performed, i.e., aquatic-based, have any additional effect on outcomes than performing exercise of another type, in this case referred to as land-based. In other words, any significant differences between groups could be due to the volume of exercise performed rather than the type. Any recommendations pertaining specifically to aquatic-based rehabilitation should thus be provided with extreme caution based on the results of the current study. A power analysis for sample size is lacking. While I appreciate the pandemic played a role in recruitment, this is no replacement for scientific rigour and the reader should at least be provided with information relating to how many participants that would have been deemed sufficient. Additionally, the current statistical analysis does not seem to be appropriate. Group means have been compared pre-intervention and post-intervention. The significant difference between groups post-intervention in favour of the experimental group, which was not evident pre-intervention, has been interpreted as the aquatic-based intervention being statistically superior to the land-based only intervention for shoulder JPS. A more appropriate statistical analysis would be instead to compare the change over time between groups. The addition of effect sizes and corrections for multiple comparisons is also recommended. Using JPS assessments is always problematic because there remains no gold standard test. It is for this reason that it would be extremely beneficial to perform reliability analysis of the particular method being used. This is lacking in the current study, which is a shame, and the results should thus be interpreted with this limitation in mind. The English language is generally understandable but would benefit greatly from proofreading. It should however be noted that the content requires a great deal of attention before proofreading is performed. In summary, while I feel the paper has merit, different statistical analyses should be performed and the interpretation of the results should consider the limitations of the study design and JPS assessment. Below are specific comments for each section of the manuscript.
Introduction
The introduction contains valuable content, but its structure and bridging between sentences/paragraphs means that it is hard to tie all aspects together in one coherent story. Further, I miss any definition of proprioception, which is important in this context and understanding how both rehabilitation programmes and assessment methods should be designed. It would, for example, be useful to understand which elements that are important to include/omit from exercises that target proprioceptive acuity. Given that the term proprioception covers a number of different senses and this particular study focuses on joint position sense, it would be interesting to know how this can be targeted. This would also allow the reader to critically review the exercises that have been implemented and described in the manuscript.
Line 41 - the word “perfect” here is unnecessary.
Line 41-42 - using a standalone sentence to open the manuscript does not seem ideal. I suggest combining with the following paragraph with the help of a bridging sentence.
Line 43 - specify neuromuscular control of the shoulder joint.
Line 56 - is a reference missing here?
Line 74 - ensure use of “JPS” rather than “joint position sense” after first use on line 69.
line 83-84 - the statement that “the visual channel (mirror therapy) and the tactile ones play an important role” is not clear. An important role in what? This highlights the need for a clear definition of proprioception. What role does vision play in proprioception according to the authors?
Line 96-101 - this section reads more like a conclusion rather than a more typical aims and hypothesis section. For example, stating “there are high statistically significant differences” is inappropriate. This may be another example where proofreading will help the manuscript, but ultimately a clear aim should be stated, with a following hypothesis.
Methods
Line 105 - Please use an alternative term to “subjects”, e.g., “participants” (check throughout manuscript).
Line 105 - it would be useful, given the later statement that the pandemic influenced the number of participants involved, to know exactly which eleven months the rehabilitation was performed. Reading further in the manuscript, I see this information is provided on line 148. I suggest including the length of the study and the exact months in line 105 only.
Line 113 - I am missing a description of how recruitment was performed.
Line 115-116 - I fully understand the issues the pandemic has brought with regard to performing research studies involving human participants. I do however miss any power calculation to give an idea of whether or not 44 participants were sufficient for the current study design. Please state the number of participants you had aimed to include with a justification.
Line 155 - “n =” should be used rather than “no =”.
Line 164 - “n =” should be used rather than “no =”.
Line 179-181 - it is stated that the “volume…was identically instrumented for both the control and the experimental group”. According to the description provided previously, the experimental group received however a total of 10 hours more rehabilitation (30 minutes, 5 times per week for four weeks). This means it is therefore impossible to know whether the differences between the groups are due to the specific type of rehabilitation exercises performed, i.e., land-based vs. hydrotherapy, or simply the greater volume of rehabilitation exercises performed by the experimental group compared to the control group. This is an unfortunate flaw in the design of the present study and limits the conclusions which can be drawn from the results.
Line 194 - a better term for “initial evaluation” is perhaps “baseline assessment”.
Line 196 - suggest “intermediate assessment” rather than “intermediate evaluation”.
Line 197 - suggest “post-intervention assessment” rather than “final evaluation”.
Line 198 - here it is stated that assessment of JPS was performed “with and without visual control”, yet the later description on lines 217-218 describes that all participants wore an opaque eye mask. Please clarify in the Methods whether or not JPS assessments were also performed with vision.
Line 207 - it is unnecessary to include the Excel file format “(*.xsls)”.
Line 208-210 - has the JPS assessment been assessed for reliability and validity? There is a lack of information regarding how the test was designed. This is important if we are to trust the outcomes.
Line 210 - it is stated that “the differences between the initial and final values” were used for JPS assessment. It is however not clear whether these were absolute values, i.e., ignoring the direction of error to provide absolute error, or whether these considered the direction of error to provide constant error. This is a very important aspect of the analysis and should be clarified. I would recommend reporting the constant error, absolute error and variable error when assessing JPS as they each provide different information.
Line 228-234 - it does not appear to be stated which statistical tests that were used to assess the nominal variables of hand dominance, sex nor fracture/dislocation type between groups.
Line 230 - it is stated that the Shapiro-Wilk test was used to determine distribution of the data series, yet it is not mentioned whether the distribution was normal or not.
Line 231 - firstly, has the student t-test been used for both intra-group comparisons and inter-group comparisons? This would seem inappropriate, as the intra-group analyses involve paired samples whereas the inter-group analyses involve independent samples. It is also not clear to me when the Welch correction has been applied, please clarify.
Line 231 - have I understood correctly that the inter-group statistical analysis has simply compared the two groups at the “initial evaluation” and then again at the “final evaluation”? If so, this is a flawed analysis. I suggest instead that the change over time for each group should be compared, i.e., the change in JPS from measurement 1 to measurement 2 should be calculated for each individual. This change over time should then be compared between groups.
Line 233 - as there is no mention of it, I assume there has been no consideration for the multiple tests? i.e., Bonferroni correction or alternative has not been applied. Could the authors please explain the justification for this.
Results
Something which is lacking from the Results section is effect sizes. Please provide these where appropriate and include in the Methods section the proposed cut-off values for interpretation.
Line 239-241 - the lack of significant differences between groups for age, height, weight and BMI is of course important in this study design. An additional factor which seems to have been overlooked is activity level. There is ample evidence that individuals with greater activity levels show better proprioceptive acuity than their peers with lower activity levels. I assume this information was not collected, which is a shame, but is encouraged should the authors continue with similar studies in the future.
Line 243-255 + Table 4 - the statistical tests and output for the variables listed in this section do not appear to have been provided.
Line 276-277 - the statement “the difference between the two rows of data (control and experimental group), is not statistically significant” is not very descriptive. Please more clearly state what the data represents, e.g., shoulder JPS was not statistically significantly different between the control and experimental groups for the baseline measurement for any of the JPS conditions. I recommend addressing this kind of content throughout the Results section. A proof-reader will be able to help with grammar, but is not likely to address scientific content and thus the authors are responsible for ensuring that this is of sufficient clarity.
Tables 5-8 - four tables for this data is unnecessary and leads to repetition, e.g., all mean values and their standard deviations are presented twice. These values could instead be condensed into one single table.
Discussion
Line 291-295 - the opening paragraph of the Discussion should pertain to the results of the current study, rather than summarising the introduction. I suggest either moving or removing the first paragraph.
Line 323-325 - the recommendation to use “hydrokinetic programs due to the mechanical and thermal benefits” cannot be drawn from the results of this study. It is equally possible that an additional 10 hours of land-based rehabilitation would have had similar effects to those of the hydrotherapy protocol. The current study design does not elucidate between these two factors.
Line 329-330 - threshold to detect passive motion (TTDPM), JPS and active movement extent discrimination (AMEDA) assess different submodalities of proprioception. This should be addressed in any discussion where comparisons are made between such assessment methods.
Line 334 - the reference for Martijn et al. is missing. Further, it seems unlikely that one study “proved” that “the results of active and passive movement tests is not statistically significant”. In fact, according to a recent meta-analysis of knee JPS tests among individuals with anterior cruciate ligament (ACL) injury (Strong A, Arumugam A, Tengman E, Röijezon U, Häger CK. Properties of Knee Joint Position Sense Tests for Anterior Cruciate Ligament Injury: A Systematic Review and Meta-analysis. Orthop J Sports Med. 2021;9(6):23259671211007878), passive tests appear to discriminate better between ACL-injured and asymptomatic knees than active tests. Although the current study relates to a different joint and population, the influence of passive/active movements on JPS assessment outcomes remains an area of research.
Conclusions
Line 360-361 - it is unnecessary to state the “investigation, interpretation, and statistical analysis of the recorded data” as this is a prerequisite to writing the conclusions of any scientific paper. It further states that the hypothesis has been confirmed, without stating the hypothesis.
Line 363 - stating the significance level is unnecessary in a conclusion. It is enough to state that differences were statistically significant. The reader can find this information in the methods section if they suspect that this may differ from the standard 0.05.
Line 364 - considering the fact that there is no power calculation for sample size and the lack of control for multiple comparisons, the significant differences between the means is not statistically validated. It may be statistically significant based on the current study design and statistical tests, but validation is not a term that should be used lightly.
Author Response
Esteemed Editor and Reviewer from Healthcare Journal,
Thank you for giving us the opportunity to submit a revised draft of the manuscript. Please see the attached point-by-point response to Reviewer #2 suggestions and comments.
Alexandra – Camelia Gliga

Reviewer 3 Report
See attached PDF in which annotations have been added.

Author Response
Esteemed Editor and Reviewer from Healthcare Journal,
Thank you for giving us the opportunity to submit a revised draft of the manuscript. Please see the attached point-by-point response to Reviewer #3 suggestions and comments.
Alexandra – Camelia Gliga

Round 2
Reviewer 2 Report
I congratulate the authors on amending their manuscript according to what I understand was extensive feedback. As a result, I feel the manuscript is now much improved. I do however feel there are certain important issues that would help to improve the manuscript further before publication, not least that the overall findings should be clarified with regard to the lack of benefit seen from the additional aquatic-based therapy. The authors also state that their manuscript has been proofread and indeed the language has improved. Many grammatical errors (although more minor than in the first version) do however also exist in the latest version of the manuscript and I suspect the person who proofread did not have English as their mother tongue. Given the amount of changes made, I provide specific comments below for each section of the latest version of the manuscript.
Abstract
The abstract does not provide sufficient background information about the participants, nor the intervention. While I understand it is impossible to include all of the information in the abstract due to word limitations, at least basic information provided succinctly is essential in the context of the study and its outcomes. I would expect information relating to whether the participants had shoulder injuries, the age and sex of the participants and the time since injury. A general outline of the rehabilitation is also beneficial. I provide suggestions here below for improvements in order to help the authors clarify their study. These are of course suggestions and I trust the authors to make the appropriate changes where they consider them to be suitable.
Line 23-24 - overuse of commas, suggest re-wording “There is limited evidence regarding the effects of aquatic-based physiotherapy on shoulder proprioception following post-traumatic injury to the joint.”
Line 24-26 - the inclusion here of “thirty-minute” is confusing as the reader is likely to believe that this is simply a one-off thirty-minute session rather than a weekly frequency. Here is also the first use of “joint position sense”, which should therefore be followed by “(JPS)”. I recommend writing the more standard “JPS” rather than “J.P.S.” throughout the manuscript. Suggest re-wording “The main aim of this study was to investigate the effects of additional aquatic-based rehabilitation to a land-based physiotherapy program on shoulder joint position sense (JPS) rehabilitation.”.
Line 26-27 - as mentioned previously, a lack of information is provided regarding the participants in the study. Suggest re-wording “Forty-four individuals (mean age xx ± xx) who had suffered a post-traumatic shoulder injury less than five months previously were pseudo-randomly allocated equally into a control group (9 females, 13 males) and experimental group (6 females, 16 males).”
Line 27-30 - suggest being more succinct, e.g., “Both groups received individualized standard land-based physiotherapy on average for 50 minutes per session, five sessions per week for four consecutive weeks. The experimental group received an additional 30 minutes of personalized aquatic-based therapy during each session.”.
Line 30-31 - here in the abstract, the terms baseline and intermediate lack definition. Suggest re-wording - “Shoulder JPS was assessed in flexion (60°), extension (25°), abduction (60°), internal rotation (35°) and external rotation (35°) positions prior, halfway during and after the intervention.”.
Line 31-33 - this statement is not clear enough. Suggest re-wording, e.g., “Shoulder JPS improved significantly for all positions for both the control group (p < 0.03) and the experimental group (p < 0.01).”.
Line 33-35 - can be written more succinctly, e.g., “No significant differences between the control group and the experimental group were found for change in shoulder JPS over time.”
Line 35-38 - caution should be taken with the word “prove” and also the emphasis on the aquatic-based therapy. Suggest re-wording, e.g., “Our results indicate that shoulder JPS can be significantly improved among individuals with post-traumatic injury to the joint through four weeks of personalized physiotherapy. The addition of aquatic-based exercises to standard land-based therapy did not, however, show significant benefits and thus cannot be recommended for the improvement of shoulder JPS based on our findings.”
Introduction
The introduction has improved greatly and the added background regarding proprioception is very useful.
Line 77-78 - what is meant by “active exercise”? Is the opposite to this passive exercise?
Line 79-82 - it is not clear whether the Casadio reference refers to the shoulder or in general, please specify.
Line 88 - when providing a reference (number 19 in this case), it is useful to provide more information about the procedures, population and study outcomes. It is hard to understand which particular CKC exercises that are being referred to, among which population, how the joint mechanoreceptors were stimulated and exactly what beneficial effect was seen.
Line 88-92 - long sentence including two references. Suggest splitting.
Line 99-101 - the benefits of hydrokinetic therapy may not be known by all the readers of this paper. I suggest providing at least a few example with primary research studies as support. This is also currently a long sentence with an overuse of commas.
Line 106-109 - is there any reason to believe that the speed, resistance and thermal effects mentioned will influence proprioception?
Line 110-115 - this is a key area of the paper which currently lacks clarity. Suggest re-wording “The aims of this study were to 1) investigate the effects of personalized physiotherapy on shoulder JPS among individuals with post-traumatic injury to the joint and 2) investigate whether additional aquatic-based therapy would have significant effects on shoulder JPS compared with only land-based rehabilitation. We hypothesized that personalized physiotherapy would significantly improve shoulder JPS and that additional aquatic therapy would improve shoulder JPS significantly more than land-based therapy alone.”
Materials and Methods
Line 128-131 - it remains unclear as to how recruitment was performed. Were these individuals contacted? Did they respond to advertisements and make first contact? Please clarify.
Line 150-151 - unnecessary to write that failure to comply with inclusion criteria resulted in exclusion. This is the whole point of inclusion criteria, i.e., if they don’t fulfil these criteria they will not be included.
Line 158 - bearing in mind the groups were split equally and characteristics were matched, then the groups were not entirely randomly distributed. This would instead be a form of pseudo-randomization. I suggest to change throughout the manuscript.
Line 200 - what is meant by the term “density” in this context? Describe how these aspects were monitored so accurately that you can be sure they were “identically instrumented”. Alternatively, use softer wording. It is hard enough to match exercise intensity even in controlled laboratory conditions.
Line 237 - “his” should be changed to “his/her”. Also, “feels” should be “felt” or even “believed”.
Line 241 - suggest “eye masks” rather than only “masks” if this is the case.
Line 246 - what is meant by the term “indifferent”?
Line 256-265 - this section is hard to understand. The first sentence, for example, states that for the comparison of absolute error means “the t-Student test was applied” and then in the following sentence “For comparison of the means, the t-Student test was used”. Why the repetition? Are these different analyses? In the following sentence, when and why was the Welch correction applied? The new analysis claims to have compared change scores, but there is no mention of these in this section. Has the Mann-Whitney test been used for these comparisons? What effect measure has been used? State how effect size is intended to be interpreted, i.e., what are your cut-off values and what is the literature you base them on? What is the post-hoc analysis that was performed?
Results
There does not seem to be any reporting relating to the intermediate assessment of shoulder JPS. Either this needs to be included and the aim of such an analysis stated clearly, or it should be stated clearly somewhere that the results this assessment were not used for analysis in the current study. It should perhaps be described somewhere what the rationale was behind the intermediate assessment as it is currently unclear.
Line 272 - the term gender is used for men/women etc. The more appropriate term here is sex, which is used for males/females.
Table 3 - unnecessary for standard deviations to be reported with three decimal places, they should instead be reported with one or two and always the same as the mean values. Further, the p value should be reported as maximum three decimal places. Also, why is the median reported? Were non-parametric tests used for the statistical comparisons? Otherwise there is no reason to report these.
Table 4 - this table can be made clearer by having as the first variable row, e.g., “Sex (female/male)” then providing proceeding values as 9/13, 40.91/59.09, 6/16, 27.27/72.73. Equally, the second row could then be “Hand dominance (left/right)” with the proceeding values as 2/20 etc. The injury information can remain as separate rows. The “Total” row at the bottom is unnecessary and should be removed.
Line 291-297 - it is not enough information to simply write that you found a statistical significance. You must also indicate the direction of difference, perhaps with the difference in means and p values/effect sizes.
Line 298-300 - what analysis has been performed here? Have you assessed whether there was a significant difference between repetition 1, 2 and 3 for each individual at each assessment, i.e., baseline, intermediate and post-intervention? This was not stated in the Methods section as a planned analysis and why state the Bonferroni correction here? What was the aim of this analysis?
Line 301-304 - because no information has been provided regarding how effect sizes were calculated, and how they should be interpreted, it is impossible to understand what these percentage values mean.
Table 6 - are these the change scores from baseline to post-intervention? If so, this should be more clearly stated in the title heading.
Discussion
Line 317 - again, the use of the term “proven” is not recommended. Please use softer phrasing, i.e., our findings indicate that…
Line 321-322 - the third finding regarding the lack of difference between groups at baseline should not be in focus here. It is of course desirable and should be reported in the results, but the main findings relate to the effects of the intervention.
Line 322-325 - smaller absoluter errors than who/when? I do not understand why this is being stated. It is important to only report whether 1) the interventions induced significant changes for both groups and 2) whether the change scores were significantly different between groups. This is then in line with the aims stated in the introduction. I suggest removing 3 and 4 and instead moving the relevant hypotheses confirmation/rejection for 1 and 2 up within the same paragraph.
Line 333-335 - again, how effort has been controlled for identically between groups is unclear. How can the “effort” of aquatic-based exercises be controlled for exactly? How can you can control the exact effort of a human being in this context?
Line 335-338 - which results are being discussed here? No significant changes were seen among the experimental group compared to the control group despite 10 extra hours of rehabilitation, so why are benefits of such training being discussed as if they have induced additional benefits?
Line 340 - the references of Brady is not discussed. How are the results similar? Provide relevant information about the reference so the reader can understand the comparisons/context here.
Line 342-343 - what does the sentence stated that “evidence….is still unclear” add to the previous sentence of “influence…is not well documented”?
Line 343-345 - again, a lack of information about the reference used regarding the population, time frame of therapy etc. Also, the comma at the end of the sentence should be removed.
Line 356-358 - the information relating to where the device was made should be included in the methods section only.
Line 358-363 - the information relating to the properties of the measurement device should be included in the methods section only.
Line 365-366 - it should be stated the specific joint and population for the range of errors. In fact, why provide a range of errors at all? It is most definitely not impossible to have errors less than 2° or greater than 7°.
Line 369-371 - what is meant by “error rates”? Do you mean just “errors”? Again, with this sentence, which joint are you referring to? Is it the shoulder? Is this for all movement directions? All angles? Has it really been proven? Are there no other potentially confounding factors? No more research needed? What does “in favour of” mean? Are errors less for passive tests than for active tests?
Line 371-374 - does this relate again to the shoulder?
Conclusion
I feel the conclusion remains slightly misleading, giving the impression that the aquatic-based therapy is beneficial to land-based therapy. It is important that this message is clearly conveyed. For example, if a clinician was to misunderstand the findings and believe that an extra 30 minutes of aquatic-based therapy was beneficial, they may include this in their treatment program. They would then be spending two and a half hours per week providing a service that does not actually provide any significant benefits. This becomes a waste of resources and an unnecessary amount of time for the patients. The concluding statement that “aquatic-based rehabilitation programs are a feasible and potential viable alternative to classical land-based rehabilitation” is not supported by the findings of this study. Instead, a more appropriate conclusion would be, e.g., “The findings of this study indicate that shoulder JPS can be significantly improved after only four weeks of personalized physiotherapy among individuals with post-traumatic injury to the joint. However, the lack of significant benefits from additional aquatic-based therapy indicates the lack of value for shoulder JPS for such exercises.”. Something that could be addressed is that only one outcome has been used in the current study, i.e., shoulder JPS (albeit use different positions), and thus effects on other outcomes are unclear. Thus, a concluding sentence could point towards future research, e.g., “Further research is required to assess whether aquatic-based therapy can provide benefits over land-based therapy on other relevant outcomes such as muscular strength and joint range of motion.”.
Author Response
Esteemed Editor and Reviewer from Healthcare Journal,
Thank you for giving us the opportunity to submit a revised draft of the manuscript. Please see the attached point-by-point response to Reviewer #2 suggestions.
Alexandra – Camelia Gliga

Reviewer 3 Report
The authors have carefully attended to all the comments raised and implemented satisfying changes to the paper. As a result the revised version of this ms has improved considerably, up to the level that I can now recommend publication.
Author Response
Esteemed Editor and Reviewer from Healthcare Journal,
We appreciate the time and effort that you and the reviewers have dedicated to providing your valuable feedback on our manuscript. We are grateful to the reviewers for their insightful comments and for the final recommendation of publication.
Alexandra – Camelia Gliga